# OpenReview forum: "What Do You See in Common? Learning Hierarchical Prototypes over Tree-of-Life to Discover Evolutionary Traits"
_ICLR.cc/2025/Conference — ICLR 2025 Poster_

### Official Review · Reviewer_z89S · 2024-11-02

**Soundness:** 3
**Presentation:** 3
**Contribution:** 2
**Rating:** 6
**Confidence:** 2

**Summary:**

This paper addresses a scientific problem: discovering evolutionary traits in biology from images. To tackle this, it proposes the Hierarchy-Aligned Commonality through Prototypical Network (HComP-Net). The primary objective is to reduce over-specific prototypes that lose common features expected to be observed in all species sharing a common ancestor. To achieve this, the approach applies contrastive, orthogonality, and discriminative losses to the prototypes, and introduces over-specificity loss and masking to mitigate over-specific prototypes.

**Strengths:**

The method details are well-demonstrated, with a clear overview and examples that make the content easy to follow.

The study presents a novel approach to improving the interpretability of prototypical networks, allowing for more accurate representation of parent-child relationships.

It enhances the specialization of deep networks applied in scientific discovery, particularly in phylogenetic analysis.

**Weaknesses:**

Two limits of experimental demonstration.
1. The primary contribution lies in incorporating parent-child relationships to reduce over-specific prototypes within the contrastive losses, while the architecture does not appear to include specific structures for establishing a prototype hierarchy.
In my understanding, proposing a hierarchical structure is a contribution of HPNet, not HCompPNet. This point should be clarified in title and method description.
2. The exclusion of certain related works leaves me a question about novelty on practical impact. For instance, PIPNet, a recent and closely related method employing self-supervised learning (2023), is not included in the comparison. It only uses HPNet from 2019. However, semantic gaps has relevance to over-specificity, and authors also mention strong motivation from PIPNet. The reason why the network is excluded needs more detailed explanation or comparison in experiments.

**Questions:**

Why does the over-specificity loss adopt a specific log-tanh loss form? Doesn’t it simply establish an arbitrary criterion for identifying overly specific prototypes? This choice requires further explanation and discussion.

---

> ### Author Response · Authors · 2024-11-21
>
> We thank the reviewer for the encouraging review and constructive feedback on our work. We have addressed the reviewer’s comments and questions in detail, and we would be happy to respond promptly if more details are required.
>
> **C1.** “The primary contribution lies in incorporating parent-child relationships to reduce over-specific prototypes within the contrastive losses, while the architecture does not appear to include specific structures for establishing a prototype hierarchy. In my understanding, proposing a hierarchical structure is a contribution of HPNet, not HCompPNet. This point should be clarified in title and method description.”
>
>
> > We agree with the reviewer that HPNet [7] was the first to introduce the concept of using a hierarchical structure for learning prototypes. In our work, we build on this foundation to address two major limitations in using HPNet and other related works for discovering hierarchical prototypes, namely the issue of finding over-specific prototypes and the absence of mechanisms for excluding such prototypes at higher levels of the tree. In the related works section, we have described how our work is related to HPnet in terms of learning prototypes at every internal node of the hierarchy. To make this point even more clear, we have now made the following modification in the introduction section to ensure that the contribution of HPnet in learning hierarchical prototypes is recognized well.
>
> > “
> Despite the success of ProtoPNet and its variants including *HPnet that first introduced the idea of learning hierarchical prototypes at every internal node of the tree*, there are three main challenges to be addressed while learning hierarchical prototypes for discovering evolutionary traits. ...
> ”
>
> **C2.** “The exclusion of certain related works leaves me a question about novelty on practical impact. For instance, PIPNet, a recent and closely related method employing self-supervised learning (2023), is not included in the comparison. It only uses HPNet from 2019. However, semantic gaps has relevance to over-specificity, and authors also mention strong motivation from PIPNet. The reason why the network is excluded needs more detailed explanation or comparison in experiments.”
> > We agree that results from PIPNet [14] can be a valuable addition to our performance comparison.  We hereby provide the results from PIPNet on all five datasets.
>
> | Model     | Bird  | Butterfly | Fish  | Spider | Turtle |
> |-----------|-------|-----------|-------|--------|--------|
> | PIPNet | 83.38 | 97.87     | 93.58 | 83.45  | 67.24  |
>
> > We have also provided comparisons with another recent interpretable approach, INTR (2024) [15].
>
> **C3.** “Why does the over-specificity loss adopt a specific log-tanh loss form? Doesn’t it simply establish an arbitrary criterion for identifying overly specific prototypes? This choice requires further explanation and discussion.”
>
>
> > The tanh term in our loss formulation is borrowed from a similar usage in PIPNet [14], although for a different goal of preventing representation collapse. The tanh term ensures that each prototype is activated at least once with respect to every descendant in a given mini batch of images. This accounts for the fact that some traits might occur frequently across the species, while others may only appear in specific subsets of images—for instance, traits observable exclusively in adult specimens but not in young ones. Therefore, tanh term is employed to avoid the criterion from being influenced by the frequency of occurrence of traits. Additionally, the negative logarithm of tanh provides a steeper gradient when tanh(x) is close to zero, helping with faster model convergence.
>
> (We have provided list of all references in the global comment)

---

> > ### Comment · Reviewer_z89S · 2024-11-26
> > **Response to Author's rebuttal**
> >
> > Thanks for the explanation and added results to reduce my concerns. I will maintain my score.

---

> > > ### Author Response · Authors · 2024-11-29
> > >
> > > Thank you very much for your valuable feedback and support!

---

### Official Review · Reviewer_PA9n · 2024-11-02

**Soundness:** 3
**Presentation:** 3
**Contribution:** 3
**Rating:** 6
**Confidence:** 3

**Summary:**

This work presents an extended approach for hierarchical prototype learning by training a neural network to discover hierarchical prototypes using structured information from a phylogenetic tree. Building upon HPNet (Haze et al., 2019), this approach, termed HComP-Net, contrasts with traditional models that use flat structures by incorporating loss functions such as over-specificity loss and discriminative loss. These functions enable HComP-Net to learn prototypes that align with the hierarchical structure of the tree, enhancing interpretability and consistency. Empirical results highlight HComP-Net’s ability to produce accurate, semantically coherent prototypes transferable to unseen species. Tested on a dataset of 190 bird species and additional organisms, the model performs better than baseline models. Additionally, HComP-Net has been shown to generate visual hypotheses on evolutionary traits, offering insight into traits across various levels of the phylogenetic tree. This research underscores the potential of interpretable representation learning using structured hierarchical prior knowledge.

**Strengths:**

Originality: The research builds on the problem initially formulated in HPNet (Haze et al., 2019), specifically addressing the challenge of over-specific prototypes. The authors introduced over-specificity and discriminative loss functions, enabling HComP-Net to learn prototypes that adhere to the hierarchical structure of a phylogenetic tree. This approach enhances interpretability and brings a fresh perspective to prototype learning.

Quality: Given the identified problem of over-specific prototypes, the authors demonstrate the efficacy of their methods across fine-grained classification tasks, also showing advancements in interpretability. The proposed model performs consistently well in these tasks, validating its quality and effectiveness.

Clarity: The paper is well-organized, with clear explanations of the background, literature, methodology, experimental setup, and results. This clarity enhances the readability and accessibility of the research, making its contributions understandable and well-supported.

Significance: This work highlights the potential of interpretable representation learning driven by structured hierarchical knowledge. By using a phylogenetic tree for guidance, the model provides insights into trait evolution and aligns closely with the biology-inspired data structure, marking a contribution at the intersection of AI and biology.

**Weaknesses:**

1) On the Need for a Separate Masking Module
The fact that an additional masking module is required suggests that the initial contributions, particularly the over-specificity and discriminative loss functions, may not fully address the issue of over-specific prototypes. This need points to a limitation in the proposed loss functions' effectiveness in preventing prototypes from becoming overly tailored to specific species. Ideally, a more robust solution would directly manage prototype specificity through the loss functions alone, reducing reliance on extra modules that could complicate the model and potentially impact interpretability or scalability.

2) On the Role of the Phylogenetic Tree in Classification
In section 5.1, the authors suggest that achieving high classification accuracy is not the primary goal. However, providing the model with additional information, like a phylogenetic tree during training and inference, could indeed aid classification performance. The tree may allow the model to leverage hierarchical relationships, which could enhance classification accuracy by using shared traits among related species. Thus, there seems to be a disconnect between the claim of not prioritizing classification accuracy and the model's design, which inherently includes information that could enhance it.

**Questions:**

1) The CNN backbone used was ConvNeXt-tiny architecture; why the fine-grained accuracy from this architecture is not included in Table 1? Will it be better/worse when compared to other methods on the Table?

2) In terms of semantically meaningful prototypes, could you discuss the possibility of obtaining similar findings using Explainable AI techniques?

---

> ### Author Response · Authors · 2024-11-21
>
> We thank the reviewer for the encouraging review and constructive feedback on our work. We have addressed the reviewer’s comments and questions in detail, and we would be happy to respond promptly if more details are required.
>
>
> **C1.** **On the Need for a Separate Masking Module**
>
> > The masking module is solving a complementary goal that is not addressed by the other components of our framework including loss terms. In particular, we use loss terms such as the over-specificity loss and discriminative loss to ensure that prototypes learned at internal nodes are common to all descendant species of that node. However, at higher levels in the tree, since the descendants can be quite diverse going far back in the process of evolution, there may be very little to no visually noticeable traits shared between them. In such cases, to ensure the learning of non-empty sets of prototypes especially at higher levels of the tree, we introduce the masking module to "slack" the constraint of dropping over-specific prototypes so that we can still perform classification at lower levels of the tree. Note that these over-specific prototypes, while useful for classification at lower levels of tree,  are not helpful in discovering evolutionary traits. Hence, we introduce a masking mechanism to identify and ignore such over-specific prototypes during the analysis of evolutionary traits. Also, since masks are just one additional parameter per prototype, they do not add significant complexity to the model.
>
> **C2.** **On the Role of the Phylogenetic Tree in Classification**
>
> > Although it is possible that providing hierarchical information can help with the classification task, the primary motivation behind using hierarchical relationships is not just to improve predictive accuracy but also focus on the interpretability of our discovered features in a format that is useful to scientists. In particular, the primary goal of our work is to learn prototypes that are representative of evolutionary traits. We can also see in our results that using hierarchical information does not always guarantee an improvement in performance. For example, we can see in Table 1 that the performance of HPnet [7] is relatively poor in comparison to other approaches despite its use of hierarchy.
>
> **C3.** **ConvNeXt-tiny performance compared to other methods on the Table**
>
> > We appreciate the reviewer’s inquiry. We did not explore multiple non-interpretable classification models including ConvNeXt-tiny [6] because our focus was on comparing our results with other interpretable and hierarchical classification methods. As we can observe in the below table, ConvNeXt-tiny, being a highly optimized architecture for feature extraction, does demonstrate strong classification performance. While we use ConvNeXt-tiny as the backbone, the observed improvements in performance of our approach compared to baselines are not solely due to the backbone. As shown in the loss ablations in Table 6 (Appendix Section D), learning hierarchical prototypes without incorporating our novel loss formulations is not so effective, even when ConvNeXt-tiny is used as the backbone.
>
>
> | Model     | Bird  | Butterfly | Fish  | Spider | Turtle |
> |-----------|-------|-----------|-------|--------|--------|
> | ConvNeXt-tiny | 84.23 | 96.28     | 91.73 | 84.05  | 69.56  |
>
>
>
> **C4.** **Possibility of obtaining similar findings using Explainable AI techniques**
>
>
> > The purpose of using explainable AI (xAI) techniques, such as CAM [3], GradCAM [4], saliency maps [5], integrated gradients [6], or LRP [7], is to clarify the classification decisions made by a classifier model. For instance, CAM offers an *object-level* interpretation by identifying the most discriminative region, possibly covering the entire object that contributes significantly to the classification outcome. The explanations are not necessarily constrained to a localized region. In contrast, prototypical approaches offer a *part-level* interpretation where every learned prototype can identify and highlight a distinct localized visual feature (or a trait). Furthermore, the aforementioned xAI techniques cannot find multiple distinct features (or parts) for a given class. In the case of prototypical approaches, every prototype learned for a class can represent a distinct feature corresponding to the same class. Therefore, for the task of identifying individual traits, prototypes are better-suited compared to other xAI approaches.
>
>
> (We have provided list of all references in the global comment)

---

> > ### Comment · Reviewer_PA9n · 2024-11-27
> >
> > Thank you, authors, for addressing my questions and concerns. I appreciate the clarity provided and find the explanations satisfactory. I will maintain my positive scores.

---

> > > ### Author Response · Authors · 2024-11-29
> > >
> > > Thank you very much for your valuable feedback and support!

---

### Official Review · Reviewer_hsuk · 2024-11-02

**Soundness:** 2
**Presentation:** 3
**Contribution:** 3
**Rating:** 6
**Confidence:** 3

**Summary:**

The paper introduces a new framework that can be used for learning biological prototypes within hierarchies while avoiding the learning
of over-specific features at internal nodes of the genetic tree. The authors perform tests with different datasets including mostly the pictures of birds, fishes and butterflies. The authors focus on the quantitative and qulitative evaluation.

**Strengths:**

- The idea presented in the paper is interesting and original. The idea is quite simple (which is a plus). I liked that the authors aimed to present a method that can help to facilitate the analysis of different species in biology.
- The paper is well-written, and also presents a lot of nice and clean graphics and examples that make the content understandable.

**Weaknesses:**

- The motivation of the paper says that there are many image datasets in the biology, so ML can be used to provide some new visual suggestions for common traits in species belonging to a common group. Nevertheless, such suggestions seem to be more important for some newly discovered species (and not the ones that are already well-known), and for such species there is a possibility of not having so many images. This can decrease the practicality of the method. Statements such as “Furthermore, HComP-Net demonstrates a unique ability to generate novel hypotheses about evolutionary traits, showcasing its potential in advancing our understanding of evolution” are too bold in my opinion.
- Another issue with a possible practical use of the method is that the proposed solution to provide semantically meaningful information requires human annotation, which can be very subjective. The authors mention this limitation in the appendix, however they do not give any solution for a mitigation.
- minor: some typos/grammatical mistakes can be found in the paper, e.g. “that are not just shared across all it descendant species but are also” -> “that are not just shared across all ITS descendant species but are also”

**Questions:**

- The paper discusses many similarities with ProtoPNet and refers to it often, e.g. “For HPnet, we used the same hyperparameter settings and training strategy as used by ProtoPNet for the CUB-200-2011 dataset”, “We follow the same training strategy as provided by ProtoPNet for the CUB-200-2011 dataset.”. Why ProtoPNet is not used in the comparative experiments (also other non-hierarchical models are used there)?
- How can the paper help to “advance our understanding of evolution”? E.g. There is a high chance that for the common species (whose pictures are available in large-scale datasets) the same features that were already used to make such a classification will be highlighted. Could the authors present a use case, in which the solution could be successfully used in practice to contribute to the understanding of evolution?
- How are the representative images (also could be perceived as prototypical images) chosen from the dataset to allow for proper explanations (some images can be taken e.g. from a non-typical angle and do not show the prototypical features well and make the explanations difficult)?
- Is it possible (and if yes, how) to use this solution in other hierarchical problems (not in the area of biology)?

---

> ### Author Response · Authors · 2024-11-21
>
> We thank the reviewer for the encouraging review and constructive feedback on our work. We have addressed the reviewer’s comments and questions in detail, and we would be happy to respond promptly if more details are required.
>
> **C1.** **Practicality of our method for newly discovered species**
>
> > We agree with the reviewer that a newly discovered species will have limited images in comparison to a well known species. However, a first step in analyzing a newly discovered species involves placing the species in the phylogeny by analyzing the synapomorphies (traits shared between species with common ancestor) it shares with existing species, before the position is further refined with gene sequencing  [1, 2, 3]. For such scenarios, an interpretable model such as ours trained on a set of existing species can be useful to identify the features the new species shares with the existing species, thereby providing an initial placement of the species in the phylogeny. This process can be done with few clean images from the new species. We also perform a similar experiment with unseen species in Section 5.1 Table 2.
>
> > Moreover, we would like to note that the number of images required in practice for training HComPNet is not significantly high. For example, the average images per species for the Bird dataset in our experiment is just close to 30, which shows that our approach can generate meaningful results even with few clean images per species.
>
>
> **C2.** **Subjectivity in the human interpretation of prototypes and ways to overcome it**
>
> > Subjectivity of interpretation is a common challenge to all approaches in interpretable AI, including prototypical networks. Subjectivity can be lessened with the inclusion of more modalities of data such as textual descriptions along with images. But this direction, to the best of our knowledge, has not been explored in any of the previous works involving prototypical networks. Our work produces prototypical parts of every species linked to evolution that are easy for biologists to visualize. However, the validation of our discovered prototypes as newly hypothesized evolutionary traits still needs further investigation by biologists.
>
> > A possible solution to reduce the subjectivity in the analysis of prototypes is to visualize the prototype in multiple training images instead of a single image (e.g., the top-k image patches closest to the prototype) so that the semantic meaning of the prototype is more apparent. A recent approach called ProtoConcepts [5] explores modifying the similarity metric to allow multiple image patches to be equally close to a given prototype. Such visualizations may help with understanding the underlying semantic concept better. We leave such an extension of our work for future research to explore.
>
>
> **C3.** **Minor: some typos/grammatical mistakes**
>
> > Thanks to the reviewer for pointing out the error. We have corrected the errors and updated the paper.
>
> **C4.** **Comparative experiment with ProtoPNet**
>
> > Since the hierarchical extension of ProtoPNet [13] (HPnet [7]) was available, comparing with HPnet seemed more reasonable. However, we hereby provide the performance of ProtoPNet on all the five datasets. We note that while ProtoPNet is performing consistently well, its hierarchical extension HPnet is unable to show consistent good performance on all datasets. On the other hand, HComP-Net shows consistently good performance (although not always better than ProtoPNet) while also learning prototypes at various levels in hierarchy. As mentioned in our global response, the main objective of our work is not to achieve the highest classification performance, but instead to discover semantically meaningful prototypes that help us explain the process of species evolution and serve as candidates in the discovery of evolutionary traits.
>
> | Model     | Bird  | Butterfly | Fish  | Spider | Turtle |
> |-----------|-------|-----------|-------|--------|--------|
> | ProtoPNet | 75.80 | 96.82     | 91.11 | 74.17  | 61.74  |
> | HPnet | 36.18 | 94.69 | 77.51 | 5.85 | 6.38 |
> | HComP-Net | 70.01 | 97.35 | 90.80 | 76.19 | 58.26 |
>
> *continued in next comment ...*

---

> ### Author Response · Authors · 2024-11-21
>
> *continuation to previous comment*
>
> **C5.** **Role in advancing our understanding of evolution and practical use case**
>
> > We agree with the reviewer that some features highlighted by HComPNet can also be identified by other models that do not consider any phylogeny. However, HComPNet’s advantage lies in assigning every discovered feature (or trait) to its respective position (ancestor node and hierarchical level) in the tree of life. For example, as shown in Figure 6, while INTR highlights the beak and neck of the western grebe species of birds, HComPNet associates these features with the correct hierarchical level, where it can be interpreted as a potential evolutionary trait. For practical use-cases, we request the reviewer to refer to Figures 5, 6, and 10–13 (Appendix). At every ancestor node, a prototype suggests a hypothesis for a potential evolutionary trait that could have developed during that particular stage of evolution. Such prototypes provide a useful starting point to further investigate how the descendant species have evolved over time.
>
> **C6.** **How prototypical images are chosen? and challenges due to non-typical angles**
> > This is certainly a challenge in working with organismal datasets because images are collected from various imaging perspectives and angles. For this reason, in most of our visualizations, we try to visualize multiple images (the top-k nearest patches to a prototype), so that we can get a better understanding of the visual features identified by a prototype. We will mention this in our discussion of limitations of our current work.
>
> **C7.** **Extending to other non-biological problems**
> > While our method is developed with the goal of identifying evolutionary traits, it can be applied without any modifications to any task where the goal is to identify commonalities among classes in-line with a known hierarchy of classes. In particular, our approach can be applied to any fine-grained datasets with known hierarchical relationships where the classes are likely to share common features with one another.
>
> (We have provided list of all references in the global comment)

---

> > ### Comment · Reviewer_hsuk · 2024-11-26
> >
> > I would like to thank the authors for their thorough answer. As I perceive the paper interesting, I will raise my score to make it positive, good luck!

---

> > > ### Author Response · Authors · 2024-11-29
> > >
> > > We are glad that you found our work interesting. Thank you very much for your valuable feedback and support!

---

### Official Review · Reviewer_hRXA · 2024-11-03

**Soundness:** 3
**Presentation:** 2
**Contribution:** 2
**Rating:** 6
**Confidence:** 3

**Summary:**

The work a novel framework called Hierarchy aligned Commonality through Prototypical Networks (HComP-Net) aimed at discovering evolutionary traits among species by learning hierarchical prototypes over the tree of life. It addresses the challenges of existing prototype-based methods that often produce over-specific prototypes at internal nodes, which can hinder the identification of common traits shared by descendant species. HComP-Net employs a unique over-specificity loss, a discriminative loss to ensure prototypes are absent in contrasting species, and a masking module to maintain classification performance. Through empirical analysis on various datasets, including birds, fishes, turtles, and butterflies, the authors demonstrate that HComP-Net effectively learns accurate, semantically consistent, and generalizable prototypes.

**Strengths:**

This paper investigates a highly intriguing task: identifying visual features preserved during species evolution. I believe this task is inherently challenging due to the limited and often insufficient quality of training data, making it difficult to obtain stable, semantically interpretable visual features. In this work, the design of the loss function and the associated explanations are intuitive and easy to understand.

**Weaknesses:**

1. The first comment is related to the choice of using visual data to identify common features in species evolution. Given the scarcity of high-quality biological images, especially in the context of vast evolutionary networks, and the inherent issues of imbalance and interference in such images, I wonder if it might be more precise to analyze common traits directly from textual descriptions or anatomical data. Could you please elaborate on the rationale behind prioritizing visual data for this task?

2. The paper introduces several loss functions aimed at ensuring the diversity and effectiveness of the learned prototypes. I would be very interested to see ablation studies on these loss functions to better understand their individual impact.

3. In Figure 4, when comparing the part consistency between HComP-Net and HPNet, different bird images are used. I am curious to know if this choice is justified and, if so, what the reasoning behind it is. Would it not be more appropriate to use the same images for a clearer comparison?

**Questions:**

Please see the weaknesses.

---

> ### Author Response · Authors · 2024-11-21
>
> We thank the reviewer for the encouraging review and constructive feedback on our work. We have addressed the reviewer's comments and questions in detail, and we would be happy to respond promptly if more details are required.
>
> **C1.** “The first comment is related to the choice of using visual data to identify common features in species evolution. Given the scarcity of high-quality biological images, especially in the context of vast evolutionary networks, and the inherent issues of imbalance and interference in such images, I wonder if it might be more precise to analyze common traits directly from textual descriptions or anatomical data. Could you please elaborate on the rationale behind prioritizing visual data for this task?”
>
> > Thanks to the reviewer for bringing up this interesting point. We agree that textual and anatomical data can also serve as a different source of information to discover evolutionary traits. However, there are two main advantages in using visual data for trait discovery compared to textual data, motivating us to work with images.
>
> > *First*, images allow machines to see what has not yet been recorded by biologists in the form of text. Since text-based annotations of traits are obtained through careful measurements of *known* parts of organisms (e.g., length of fins or color of beaks), they may not contain information about novel trait variations that are yet *unknown* to biologists. For example, there are subtle variations in traits such as the precise shape of leaf venations in plants or wing morphology in insects [4] that are difficult to capture in text-based descriptions but can be observed in images (we show some newly discovered traits of wing morphology on the Butterfly dataset in Figures 10, 11, and 14 to 17). By leveraging images, we can enable machines to see what biologists do not see, leading to the discovery of novel fine-grained traits linked to evolution which is the motivation for this work.
>
> > *Second*, the scale of image-based datasets in organismal biology is vastly greater than that of expert-curated text or anatomical datasets of species. Images are increasingly being considered as the “currency” for documenting biodiversity, with repositories containing millions of images of biological specimens collected by scientists in field museums or captured by drones, camera traps, or tourists posting photos on social media. On the other hand, preparing expert-curated text or anatomical data of species involve subjective and labor-intensive operations hindering rapid scientific advancements. This is especially true for newly discovered species for which detailed textual descriptions of traits may not be yet available. For such species, a vision based approach such as ours can be useful as a first approach in placing the species on the phylogeny based on shared visual traits with other species. This initial placement can then be further refined through gene sequencing [1, 2, 3].
>
> > Therefore, we believe that despite the limitations of vision modality (such as imbalance and interference), tools for analyzing visual data can serve a unique purpose in the analysis of species, especially when textual or morphological data are not available in abundance.
>
> **C2.** “The paper introduces several loss functions aimed at ensuring the diversity and effectiveness of the learned prototypes. I would be very interested to see ablation studies on these loss functions to better understand their individual impact.”
>
> > The ablation for key loss terms in our approach have been provided in Appendix Section D Table 6. We hope that it provides clarity on the impact of each individual loss term. We will be happy to provide analyses of additional ablations of our approach based on the reviewer's suggestions.
>
> **C3.** “In Figure 4, when comparing the part consistency between HComP-Net and HPNet, different bird images are used. I am curious to know if this choice is justified and, if so, what the reasoning behind it is. Would it not be more appropriate to use the same images for a clearer comparison?”
>
>
> > In order to have a fair comparison between the prototypes for part consistency, we have taken the top-3 closest images to the respective prototypes for visualization. Therefore, we see the set of images to differ, since top-3 closest image patches can vary between prototypes.
>
> (We have provided list of all references in the global comment)

---

### Official Review · Reviewer_ta8u · 2024-11-04

**Soundness:** 2
**Presentation:** 3
**Contribution:** 3
**Rating:** 6
**Confidence:** 4

**Summary:**

This paper applies deep learning techniques to uncover evolutionary traits in biological data. It leverages contrastive and orthogonality losses to facilitate hierarchical prototype learning. Additionally, the paper introduces over-specificity and discriminative losses to guide and constrain model training. The proposed method demonstrates improved performance over baseline methods across multiple benchmark datasets.

**Strengths:**

1. The motivation is interesting and well-justified. The hierarchical structure of biological data presents significant challenges for distinguishing species and identifying evolutionary traits.
2. The techniques employed in HComP-Net appear technically feasible. The use of contrastive loss for learning clustered features has proven effective in self-supervised learning, while orthogonality loss helps capture diverse features.
3. The visualization results are clear and impressive.
4. The paper is well-structured and easy to follow.

**Weaknesses:**

1. It seems that the over-specificity and discriminative losses play opposing roles in the direction of model optimization, which raises the question of whether these two losses might interact and lead to abnormal model convergence. It would be beneficial if the authors could provide some theoretical or experimental analysis on this issue.
2. This is a minor point, but the overall method appears to be a combination of multiple techniques, making the flowchart somewhat complex and redundant.

**Questions:**

Please kindly refer the the weaknesses.

---

> ### Author Response · Authors · 2024-11-21
>
> We thank the reviewer for the encouraging review and constructive feedback on our work. We have addressed the reviewers comments and questions in detail, and we would be happy to respond promptly if more details are required.
>
> (Please find the references in the global comment)
>
> **C1.** “It seems that the over-specificity and discriminative losses play opposing roles in the direction of model optimization, which raises the question of whether these two losses might interact and lead to abnormal model convergence. It would be beneficial if the authors could provide some theoretical or experimental analysis on this issue.”
>
> > We would like to clarify that the over-specificity and discriminative losses are not playing an opposing role to each other. For clarity, the losses can also be interpreted as clustering and separation losses generally used in clustering algorithms. The over-specificity loss encourages prototypes to be equally close to all species within the positive set (descendant species), while the discriminative loss ensures that these prototypes are separated from species in the negative set (non-descendant species). Together, these losses work to align the learned prototypes with the phylogenetic structure, ensuring they are likely to represent potential evolutionary traits.
>
> **C2.** “This is a minor point, but the overall method appears to be a combination of multiple techniques, making the flowchart somewhat complex and redundant.”
>
> > Thanks to the reviewer for the feedback. Our aim with the schematic illustration was to provide a clear and comprehensive overview of the various components of our approach. Specifically, we intended to highlight where our  key contributions, such as the novel loss formulations and the masking module, are integrated into the architecture. Additionally, we chose to depict the prototypes, classification layer, and masking module, repeated for each node, to emphasize that these layers are learned for each internal node in the hierarchy. However, we appreciate the feedback and would be happy to consider any specific suggestion the reviewer may have in mind for simplifying the schematic while maintaining clarity and completeness.

---

> > ### Comment · Reviewer_ta8u · 2024-11-25
> >
> > Thanks for the authors' responses. My concerns are addressed, and I'll keep my positive rating.

---

> > > ### Author Response · Authors · 2024-11-25
> > >
> > > Thank you very much for your feedback and support!

---

### Author Response · Authors · 2024-11-21
**Global Response to All Review Comments**

We sincerely thank all the reviewers for providing constructive feedback. We are encouraged that the reviewers found our work:
1. Original and interesting (hsuk, PA9n, z89S)
2. Well written with impressive visualizations (ta8u, z89S, hsuk)
3. Impactful in phylogenetic analysis (z89S, PA9n)
4. Intuitive and technically feasible (ta8u, hRXA)

Before addressing each of the reviewer's comments individually, we would like to address two general comments shared in multiple reviews.

The first general comment is on including **comparisons with more baselines** including ProtoPNet [13] (asked by Reviewer hsuk), PIPNet [14]  (asked by Reviewer z89S), and ConvNeXt-tiny (asked by reviewer PA9n). To address this comment, we have included comparisons with additional baselines as summarized in the following table.

| Model       | Bird  | Butterfly | Fish  | Spider | Turtle |
|-------------|-------|-----------|-------|--------|--------|
| PIPNet      | 83.38 | 97.87     | 93.58 | 83.45  | 67.24  |
| ConvNeXt-tiny  | 84.23 | 96.28     | 91.73 | 84.05  | 69.56  |
| ProtoPNet   | 75.80 | 96.82     | 91.11 | 74.17  | 61.74  |

The second general comment is on the **primary objective of our work** (hsuk and z89S). We would like to re-emphasize that the main motivation for building HComP-Net is to learn semantically meaningful prototypes of a hierarchy of classes (e.g., the tree of life) that helps in *explaining* what are the common features (or traits) that are shared by multiple classes. Our focus is on discovering hierarchical prototypes that are explainable and scientifically useful, e.g., in the target application of discovering evolutionary traits. While the use of hierarchy also helps HComP-Net in improving classification accuracy as demonstrated in our results in Table 1, achieving the best classification accuracy that beats all state-of-the-art (SOTA) classification methods is not our end-goal, unless the learned features are explainable and scientifically useful for discovering hierarchical prototypes. We demonstrate the ability of our approach in discovering semantically meaningful hierarchical prototypes in comparison with baseline hierarchical methods both qualitatively and quantitatively in Section 5.3 of the paper.

We hope that our responses to the individual review comments provided below address the main concerns of the reviewers. If we missed any detail, we will be happy to provide more clarifications during the discussion period. If our responses have adequately addressed your concerns, we kindly request that you consider revising your scores. Thank you very much for your time and effort.

---

> ### Author Response · Authors · 2024-11-21
> **References Used in the Response**
>
> The set of all reference used throughout our responses have been listed here.
>
> [1] Nathan D. Smith, Alan H. Turner, Morphology's Role in Phylogeny Reconstruction: Perspectives from Paleontology, Systematic Biology, Volume 54, Issue 1, February 2005, Pages 166–173, https://doi.org/10.1080/10635150590906000
>
> [2] Larson, A. (1998). The comparison of morphological and molecular data in phylogenetic systematics. In: DeSalle, R., Schierwater, B. (eds) Molecular Approaches to Ecology and Evolution. Birkhäuser, Basel. https://doi.org/10.1007/978-3-0348-8948-3_15
>
> [3] Per G P Ericson, Yanhua Qu, An evaluation of the usefulness of morphological characters to infer higher-level relationships in birds by mapping them to a molecular phylogeny, Biological Journal of the Linnean Society, 2024;, blae070, https://doi.org/10.1093/biolinnean/blae070
>
> [4] Vasyl Alba, James E Carthew, Richard W Carthew, Madhav Mani (2021) Global constraints within the developmental program of the Drosophila wing eLife 10:e66750
>
> [5] Ma, C., Zhao, B., Chen, C. and Rudin, C., 2024. This looks like those: Illuminating prototypical concepts using multiple visualizations. Advances in Neural Information Processing Systems, 36.
>
> [6] Liu, Z., Mao, H., Wu, C.Y., Feichtenhofer, C., Darrell, T. and Xie, S., 2022. A convnet for the 2020s. In Proceedings of the IEEE/CVF conference on computer vision and pattern recognition (pp. 11976-11986).
>
> [7] Hase, P., Chen, C., Li, O. and Rudin, C., 2019, October. Interpretable image recognition with hierarchical prototypes. In Proceedings of the AAAI Conference on Human Computation and Crowdsourcing (Vol. 7, pp. 32-40).
>
> [8] Zhou, Bolei, et al. "Learning deep features for discriminative localization." Proceedings of the IEEE conference on computer vision and pattern recognition. 2016.
>
> [9] Selvaraju, Ramprasaath R., et al. "Grad-cam: Visual explanations from deep networks via gradient-based localization." Proceedings of the IEEE international conference on computer vision. 2017.
>
> [10] Simonyan, Karen. "Deep inside convolutional networks: Visualising image classification models and saliency maps." arXiv preprint arXiv:1312.6034 (2013).
>
>
> [11] Sundararajan, Mukund, Ankur Taly, and Qiqi Yan. "Axiomatic attribution for deep networks." International conference on machine learning. PMLR, 2017.
>
> [12] Binder, Alexander, et al. "Layer-wise relevance propagation for neural networks with local renormalization layers." Artificial Neural Networks and Machine Learning–ICANN 2016: 25th International Conference on Artificial Neural Networks, Barcelona, Spain, September 6-9, 2016, Proceedings, Part II 25. Springer International Publishing, 2016.
>
> [13] Chen, C., Li, O., Tao, D., Barnett, A., Rudin, C. and Su, J.K., 2019. This looks like that: deep learning for interpretable image recognition. Advances in neural information processing systems, 32.
>
> [14] Nauta, M., Schlötterer, J., Van Keulen, M. and Seifert, C., 2023. Pip-net: Patch-based intuitive prototypes for interpretable image classification. In Proceedings of the IEEE/CVF Conference on Computer Vision and Pattern Recognition (pp. 2744-2753).
>
> [15] Paul, D., Chowdhury, A., Xiong, X., Chang, F.J., Carlyn, D., Stevens, S., Provost, K.L., Karpatne, A., Carstens, B., Rubenstein, D. and Stewart, C., 2023. A simple interpretable transformer for fine-grained image classification and analysis. arXiv preprint arXiv:2311.04157.

---

> ### Author Response · Authors · 2024-11-25
> **Request for Final Reviewer Queries and Score Updates**
>
> We sincerely appreciate the insightful comments and valuable feedback provided by our reviewers. As we near the conclusion of the discussion period, we would like to extend an invitation for any further questions or clarifications. If our responses have adequately addressed your concerns, we kindly request that you consider revising your scores and confidence accordingly. Thank you for your time and dedication to this review process.

---

> ### Author Response · Authors · 2024-12-04
> **Final summarization of major reviewer concerns and responses**
>
> We would like to once again thank the reviewers for their time and effort in the review process. We hereby provide a concise summary of some of the key concerns raised by the reviewers and our responses to clarify them. Although we have covered only the major concerns here, complete set of queries and the detailed responses can be found under the respective reviewer comments.
>
> **C1. Primary objective of our work** (Reviewers hsuk and z89S)
> > As we have discussed in the global comment, our primary objective is to learn semantically meaningful prototypes representing common features (or traits) that are shared by multiple species in line with the phylogeny, thereby discovering potential evolutionary traits. Therefore, our motive is not necessarily to achieve the best classification accuracy that beats all state-of-the-art (SOTA) classification methods. We demonstrate the ability of our approach in discovering semantically meaningful hierarchical prototypes in comparison with baseline hierarchical methods both qualitatively and quantitatively in **Section 5.3 and 5.4** of the paper.
>
> **C2. Regarding the choice of using visual data for finding common traits** (Reviewer hRXA)
> > We would like to note two advantages in working with visual data. *Firstly*, images can capture subtle variations in traits such as the precise shape of leaf venations in plants or wing morphology in insects [4] that are difficult to capture in text-based descriptions. *Secondly*, with the growing availability of image datasets in organismal biology, images are easier to obtain compared to expert-curated text or anatomical data. This is especially true for newly discovered species for which detailed textual descriptions of traits may not be yet available. For such cases, vision-based approaches can be useful in performing initial placement of the species in the phylogeny. Therefore, vision-based tools can serve a unique purpose in the analysis of species, especially when textual or morphological data are not available in abundance.
>
> **C3. Subjectivity in the human interpretation of prototypes and ways to overcome it** (Reviewer hsuk)
> > Subjectivity of interpretation is a common challenge to all approaches in interpretable AI, including prototypical networks.  A possible solution to reduce the subjectivity is to visualize the prototype in multiple training images so that the semantic meaning of the prototype is more apparent. For this reason, we visualize top-k images per prototype in most of our visualizations instead of a single image.
>
> **C4. Practicality of our method for newly discovered species with less  available images** (Reviewer hsuk)
> > Newly discovered species are likely to have less available images in comparison to well known species. However, a first step involved in placing a new species on the phylogeny is analyzing synapomorphies (traits shared between species with common ancestor), before the position is further refined with gene sequencing [1, 2, 3]. Our interpretable model can serve as a tool for performing such initial placement. We also perform a similar experiment with unseen species in **Section 5.2 Table 2**. Moreover, in our experiments we work with Bird dataset where the average number of images per class is only about 30, which shows that our approach can generate meaningful results even with few clean images per species.
>
> **C5. On the Need for a separate masking module in addition to the loss terms** (Reviewer PA9n)
> > Masking module serves a complementary role that is not addressed by the loss terms. Over-specificity and discriminative losses ensure the prototypes learned at internal nodes are common to all descendant species of that node. However, at higher levels of tree, the descendants are quite diverse and therefore have little to no visually noticeable shared traits between them. At such higher levels, over-specific prototypes are required to ensure a non-empty set of prototypes to meet the classification objective. However, since they do not represent potential evolutionary traits, masking module helps us to identify and exclude such over-specific prototypes during the analysis of evolutionary traits.
>
> (We have provided list of all references in the global comment)

---

### Comment · Area_Chair_FzGH · 2024-11-25
**Please enage in the discussion**

Dear all,

Many thanks to the reviewers for their constructive reviews and the authors for their detailed responses.

Please use the next ~2 days to discuss any remaining queries as the discussion period is about to close.

Thank you.

Regards,

AC

---

### Meta-Review · Area_Chair_FzGH · 2024-12-19

**Metareview:**

HComP-Net (Hierarchy aligned Commonality through Prototypical Networks) is a new machine learning framework that discovers evolutionary traits in species by analysing images and learning hierarchical prototypes aligned with phylogenetic trees. The system's key contribution is its ability to identify visual features shared by species with common ancestors while avoiding "over-specific" prototypes that only apply to some descendant species.

The framework introduces three main technical contributions: an over-specificity loss to ensure prototypes are genuinely common across descendant species, a discriminative loss to ensure prototypes are absent in contrasting species groups, and a masking module to identify and exclude over-specific prototypes without impacting classification performance.

Tested on datasets of birds, fish, butterflies, spiders, and turtles, HComP-Net demonstrates better semantic consistency in identified prototypes compared to baselines and shows promise in generalising to unseen species. While its classification accuracy is somewhat lower than non-interpretable models, this trade-off enables clear interpretability and the generation of testable hypotheses about evolutionary traits.

In terms of some of its limitations, the system works best with high-quality images and is limited to analysing local, visually observable traits rather than global features. Despite this, HComP-Net represents a neat approach to automatically discover and analyse evolutionary traits from image data.

In a nutshell, its contribution is bridging computer vision and evolutionary biology by providing a tool that can automatically identify and analyze traits that may have evolved over time, potentially accelerating scientific discovery in evolutionary biology.

However, it might be seen as talking to a narrow audience given experimental setup, but it should be acknowledged that it has been submitted to the applications to physical systems track.

**Additional Comments On Reviewer Discussion:**

All five reviewers have unanimously voted to accept this paper, while recognising the very targeted nature of this paper in terms of technical novelty, comparisons with other baselines, and the overall usefulness in other domains.
The authors engaged well with the reviewers and addressed most of the issues identified.
In balance, everyone agrees that despite the limitations, this paper is worth publishing.

---

### Decision · Program_Chairs · 2025-01-22

Accept (Poster)